# 'It's like being in a war with an invisible enemy': A document analysis of bereavement due to COVID-19 in UK newspapers

Ryann Sowden[1], Erica Borgstrom[2], Lucy E. Selman[1]*

1 Palliative and End of Life Care Research Group, Population Health Sciences, Bristol Medical School, Bristol, United Kingdom, 2 School of Health, Wellbeing and Social Care, The Open University, Milton Keynes, United Kingdom

* lucy.selman@bristol.ac.uk

**Data Availability Statement:** All relevant data are available in figshare:10.6084/m9.figshare. 14074547.

## Abstract

The COVID-19 pandemic has been followed intensely by the global news media, with deaths and bereavement a major focus. The media reflect and reinforce cultural conventions and sense-making, offering a lens which shapes personal experiences and attitudes. How COVID-19 bereavement is reported therefore has important societal implications. We aimed to explore the reportage and portrayal of COVID-19 related bereavement in the top seven most-read British online newspapers during two week-long periods in March and April 2020. We conducted a qualitative document analysis of all articles that described grief or bereavement after a death from COVID-19. Analysis of 55 articles was informed by critical discourse analysis and Terror Management Theory, which describes a psychological conflict arising between the realisation that death is inevitable and largely unpredictable and the human need for self-preservation. We identified three main narratives: (1) fear of an uncontrollable, unknown new virus and its uncertain consequences—associated with sensationalist language and a sense of helplessness and confusion; (2) managing uncertainty and fear via prediction of the future and calls for behaviour change, associated with use of war metaphors; and (3) mourning and loss narratives that paid respect to the deceased and gave voice to grief, associated with euphemistic or glorifying language ('passed away', 'heroes'). Accounts of death and grief were largely homogenous, with bereavement due to COVID-19 presented as a series of tragedies, and there was limited practical advice about what to do if a loved one became seriously ill or died. Reporting reflected the tension between focusing on existential threat and the need to retreat from or attempt to control that threat. While the impact of this reporting on the public is unknown, a more nuanced approach is recommended to better support those bereaved by COVID-19.

## Introduction

The World Health Organization declared COVID-19 a pandemic on 11th March, 2020. The disease has been reported internationally with most governments implementing infection

**Funding:** The authors received no specific funding for this work, however LES and RS are funded by a National Institute of Health Research (UK) Career Development Fellowship to LES. The funders had no role in study design, data collection and analysis, decision to publish, or preparation of the manuscript.

**Competing interests:** The authors have declared that no competing interests exist.

control measures [1] and many countries facing mass bereavement. At the time of writing, COVID-19 has killed over 2.3 million people worldwide [2]; over 120,000 in the UK [3]. While there are likely to be large variations in how bereavement affects a population, recent research in the USA estimates that each death from COVID-19 leaves 9 people bereaved [4]. Using this metric, an estimated 1.08 million people in the UK are currently grieving a COVID-19 death. Death has thus been brought to the forefront of public consciousness, with members of the public confronted with the prospect of themselves or someone they know dying suddenly.

Though often described as a societal taboo or private affair in British culture, death remains ever-present in the public realm of news and media [5]. As Walter has said: "Death sells newspapers. . . by the bucketload; far from marginal, it is integral to the capitalist economy of western media" [6 p.12]. In an age of online newspapers and social networks, media play a key role in cultural reflection and collective sense-making [7], highlighting and enforcing cultural conventions and attitudes, especially at times of health emergency [8]. Media tropes and techniques influence the public: creating a lens which shapes experiences and attitudes, highlighting socio-cultural fears and anxieties. During the COVID-19 pandemic, the media have played a central role in informing the public and cultural sense-making, with newspaper articles omnipresent across social media and reported on television and radio [9].

The role of the media in reporting pandemics and other mass bereavement or emergency events has been previously explored. Basnyat and Lee [10] discuss how the use of journalistic frames such as war metaphors during the H1N1 pandemic constructed a narrative of a need to fight and win against disease. Analysing broadcast media during the Ebola outbreak, Kott and Limaye [8] found that radio shows used multiple voices and frames which could confuse viewers and affect the processing of information. Media analysis can therefore illuminate perspectives on novel phenomena and social attitudes [5], including in relation to death and grief [7].

UK online newspaper readership increased by 6.6 million in the first quarter of 2020 as a result of the COVID-19 pandemic [11]. The UK media's focus on COVID-19 sharpened following the first recorded deaths of UK citizens in February 2020, and came to dominate the news agenda following the first in-country deaths in mid-March. Evidence from previous epidemics suggests increased coverage and sensational, insensitive or inaccurate reporting can contribute to public fear, moral panic and individual grief [12, 13]. Although there has been some attention to the media's coverage of COVID-19 [14], only one article has explored the reporting of deaths. Focusing on Bangladeshi reports linking individual deaths to COVID-19, Goni [15] found this to be a stigmatising practice which resulted in bereaved people facing barriers to bury their loved ones. Goni therefore recommends that journalists provide factual, objective coverage of death during the pandemic [15].

In this study we aimed to explore reportage of COVID-19 related death, grief and bereavement in British online newspapers during a one-month period in spring 2020. After initial analysis, we used the lens of Terror Management Theory (TMT) to interpret the findings. TMT describes the psychological conflict between the human need for self-preservation and the realisation that death is inevitable and largely unpredictable [16]. Terror is managed by taking increased comfort in one's cultural worldview (such as religious or nationalistic identities), minimising uncertainty by "weaving the individual more securely into a meaningful cultural fabric" [17 p.307] and providing a sense of 'symbolic immortality' through which one lives on after death [9]. Here we describe the narratives surrounding COVID-19 related death and bereavement in UK online newspapers, how these narratives enact the tension between self-preservation and mortality, and the insight they provide into specific cultural world-views.

## Methods

We conducted a document analysis of online newspaper articles in the public domain. We focused on online newspaper articles—a readily publicly available source of discursive data that represents a socially influential form of discourse (widely discussed in person, on social media, and on broadcast media) with cultural influence beyond its own medium.

Our approach is based on qualitative document analysis [18] and informed by both critical discourse analysis, which addresses both the explicit and implicit understandings within a discourse [19], and Terror Management Theory (TMT) [16]. This analytical approach enables us to describe implicit and explicit sense-making newspaper narratives in the context of COVID-19, and the wider implications of these perspectives.

### Data collection

Online UK newspapers and their Sunday counterparts were ranked according to their readership numbers [20]. The top seven most-read papers were selected for inclusion, representing a range of political perspectives, readership demographics and newspaper types: former broadsheets (The Guardian, The Telegraph and The Times); a middle market newspaper (The Daily Mail); and tabloids (The Mirror, The Sun and The Metro) [21]. The Daily Mail and The Telegraph have the largest readership among people aged over 65, while The Guardian, The Sun and Metro have the largest readership among those aged 16–24 [22]. The Mirror and The Times have a similar readership rates across age groups [22]. The Metro has the largest readership amongst people from an ethnic minority background [22].

The time period for the search was four weeks—18/03/2020 to 14/04/2020—selected to represent a time when deaths were becoming widely reported. It included a period before the UK government imposed 'lockdown,' the beginning of the first lockdown, and time before the lockdown extension.

Searches were completed through each newspaper's search function using key words: grief, bereavement, COVID-19 (and the variants covid-19 and sars-cov-2), coronavirus, pandemic. These were completed as Boolean searches, with grief or bereavement combined with the disease. Search terms were limited to the English language. Search results were screened; articles were included if they described grief and bereavement following a human death from COVID-19 during the pandemic. Articles were excluded if they were: publications from countries other than the UK; articles written for a non-UK audience (e.g. the 'Australia' section of The Daily Mail); articles in which key words occurred but not in relation to human death (e.g. widespread reporting of the Queen's speech, which used the term 'grief'); obituaries; articles discussing bereavement and grief occurring from non-COVID-19 deaths; and articles discussing grief unrelated to death (e.g. grieving loss of social contact), or a non-human death (e.g. of a pet) as this was outside the focus of the study.

The search yielded high numbers of relevant results (Table 1), which we organised into weekly groups. Fig 1 shows a flowchart outlining the screening process. Due to the high numbers of articles, to explore changes over time, and to encompass key points of the COVID-19 UK timeline, we selected the first and last week of this month-long period for analysis. The first week (Week A: 18/03/2020-24/03/2020) covered the UK's transition into the first lockdown; the fourth week (Week B: 08/04/2020-14/04/2020) occurred at the first peak of the pandemic. The Daily Mail had the most articles across both weeks (week A = 8; week B = 20), representing just under half of all the articles included. S1 Table shows the number of included articles by date, contextualised with total deaths to date and key events. Data extracts are tagged with a unique ID code (Table 1), referring to the publication, week category (A = 18/03/2020-24/03/2020; B = 08/04/2020-14/04/2020) and article number; for

Table 1. Search results by publication and date range.

| | The Guardian (G.) | The Daily Mail (DMa.) | The Telegraph (Te.) | The Mirror (TMi.) | The Sun (S.) | The Times (Ti.) | The Metro (Me.) | Total |
|---|---|---|---|---|---|---|---|---|
| **18/03/2020–24/03/2020 (week 1)** | **3** | **8** | **1** | **2** | **1** | **2** | **2** | 19[a] |
| 25/03/2020–31/03//2020 | 4 | 5 | 1 | 3 | 4 | 0 | 3 | 20 |
| 01/04/2020-07/04/2020 | 3 | 7 | 5 | 8 | 8 | 3 | 2 | 36 |
| **08/04/2020–14/04/2020 (week 4)** | **1** | **20** | **2** | **2** | **5** | **3** | **3** | 36[a] |
| Total by publication title | 11 | 40 | 9 | 15 | 18 | 8 | 10 | 111 |

[a]Included in the analysis.

example, S.A.13 refers to an article in The Sun, published in the first week, and is the 13th article included.

## Data analysis

We followed the principles of qualitative document analysis [18] querying how behaviour and events are placed in context, identifying what themes, frames, and discourse are presented [18]. Document analysis can reveal the meanings of acts and how these are framed, illustrating key stylistic differences between different publications. The frames used by the media suggest what is considered relevant and irrelevant in relation to a phenomena [18]. In this study, the act of interest is bereavement and grief due to COVID-19; our analysis describes and interprets the main frames and themes in how this phenomenon was presented in mainstream UK newspapers.

While qualitative document analysis is an established research method in explorations of media representation, there is no single accepted methodological approach [23, 24]. We drew on Altheide's [18] emergent/ethnographic analysis methods: deep immersion in the data and asking questions about the organisation, production and consequences of the content. Our analysis began with conventional content analysis [25]. Coding focused on manifest themes (types and aims of the articles, how the subject is framed), latent themes (related to the content, use of metaphors/symbols, contradictions and confusions), and any changes in trends and emphases over time and across publications.

To develop a coding framework, three varied and rich articles were selected by RS from the Week A dataset. Each co-author independently read these articles, making notes on latent and manifest themes. The coding and notes were discussed as a group, and organised to develop a hierarchical coding frame of defined themes and sub-themes for the first week (LES). This process was repeated for the Week B dataset. These coding frames were combined, refined and expanded to capture the themes and salient issues from both weeks. RS applied the coding frame to all included articles, with regular meetings with LES and EB to discuss narrative formation within articles, themes within and across each week, and social theories that could help us understand the discourses. We reflected on our re-reading of the articles as the pandemic unfolded and how our interpretations could shift over time due to this. Additional emergent themes (not identified in the initial coding framework) were discussed, added as needed, and applied consistently across the dataset by RS. NVivo 12 was used for data management.

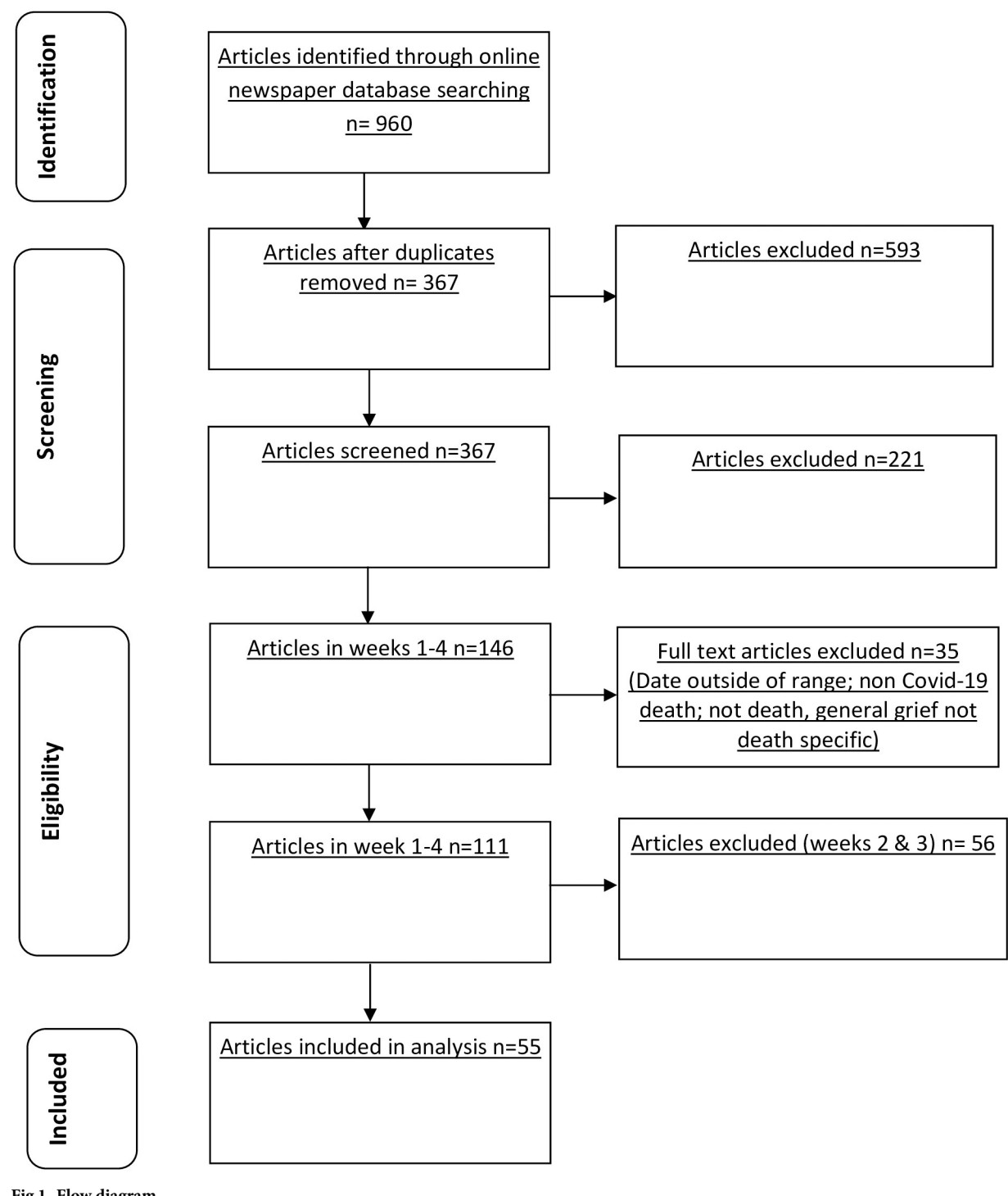

**Fig 1. Flow diagram.**

In later stages of the analysis we used TMT as an interpretative lens to explore media sense-making of COVID-related grief. The use of appropriate theoretical models in qualitative research can aid and enhance in-depth analysis [26] especially when studying complex phenomena [27, 28]. We selected TMT as it complemented many codes identified in the initial

content analysis (for example, related to the effects of funeral restrictions on the bereaved and the evident fear and control in reportage), and provides a useful analytical perspective on the themes and tropes related to newspapers' sense-making during the COVID-19 pandemic. TMT has been applied across disciplines and found to be applicable in different religious and cultural groups [29, 30] and in disasters and previous pandemics [31–33]. TMT has been used to explain differences in health-related behaviour during the pandemic [34], but has not previously been used to make sense of media reporting of bereavement during COVID-19.

## Results

Searches resulted in 55 articles for inclusion in the analysis: 19 from Week A and 36 from Week B. Many of the increased number of articles in Week B focused on the deaths of individuals, perhaps reflecting growing numbers of UK deaths and a growing awareness of the relevance of grief and bereavement caused by the pandemic. Articles in both weeks comprised news reports and features, varying in length.

We noted differences between publications, which may reflect differences in form and content between publications. The Guardian and Telegraph focused on funerals, The Daily Mail discussed pessimistic possible outcomes both in the UK and overseas, The Times focused on reporting deaths, while The Mirror and The Sun focused on individual tragedies. The Metro's focus was broader, including two articles with an unusually positive, practical tone (Me.A.16 and Me.B.46). News reports primarily focused on death counts and policy changes; features primarily focused on funerals, celebrity deaths and advice.

We identified three narratives around COVID-19 death and bereavement: fear of an uncontrollable, unknown new virus and its uncertain consequences; managing uncertainty and fear through prediction and calls to action; and mourning and loss. Each narrative was signified through specific language, metaphor and associated emotion(s). The three narratives often intersected within articles. We identified latent themes and gaps in each of the narratives presented, particularly in relation to positive perspectives and advice.

### Fear of an uncontrollable, unknown new virus and its consequences

This narrative presented COVID-19 as an uncontrollable, unpredictable new virus about which little was known. COVID-related death and destruction was portrayed as a universal possibility, with sensationalist headlines emphasising its lethal nature and risks. Presentation of this narrative both reflected and stoked public fear of the disease:

S.B.44: DES-PPE-RATE TIMES Terrified NHS staff beg for PPE as eight more health workers die from coronavirus

DMa.A.19: Coronavirus victims are dying in quarantine and left to be buried or cremated away from grieving relatives, with the crisis having already claimed 9,000 lives.

The uncontrollable and unpredictable nature of the virus was conveyed through passive language in relation to the people it affected. The deceased were "struck down or "succumbed" to the virus, which "claimed the lives" of thousands, leaving the bereaved "grief-stricken." Descriptions of a "mounting" death toll and a "surge" in fatalities suggested an inability to stop the deadly virus.

The potential for human loss was fundamental to COVID-19's fear-associations. Tragic personal stories and case studies humanised the 'victims' of COVID-19 and underscored the threat's immediately. Stories often highlighted incidents in which COVID-19 affected whole

families or killed children and people otherwise expected to recover. Victims were often described as previously healthy, with underlying health conditions (such as diabetes) only reported in later paragraphs. Articles often used bereaved people's voices to warn of compounded loss: first the loss of loved ones, then the loss of the expected grieving rituals due to pandemic response measures. This conveyed a 'double' sense of terror: the terror of death itself alongside disruption to cultural or religious customs that help reduce existential anxiety. Particularly tragic cases were highlighted in article headlines or subheadings alongside the daily death count reported regularly across media outlets:

Te.B.31: Deaths in the UK rise by 917 as 11-year-old has died from the virus

Articles linked COVID-related fear to the rapidly-changing nature of the pandemic, with uncertainty adding to an underlying panic. Clear and direct descriptions of what could happen at each stage of a person being taken ill were rare. A latent theme identified here was a lack of clarity in the rules governing hospital visits, compounded by variation in the interpretation of such guidance. Language used in these descriptions was typically vague (for example, describing families as "unable" to visit hospitals), without reporting the specific details behind such constraints (be it closed hospitals, shielding and self-isolation requirements, or travel restrictions). This journalistic perspective emphasised problems and barriers over solutions and support, playing into fears regarding a lack of control during the hospitalisation of loved ones.

Though funeral restrictions were published as recommendations rather than rulings on 18[th] March 2020 [35], there was a lack of clarity about their status and content in newspaper reporting. Fear narratives were linked to sensationalism: a common technique in newspapers, used to drive and engage readership while provoking outrage. The word 'ban,' for example, was used repeatedly to refer to funeral restrictions. If such sensationalism is accepted by readers, it may affect cultural perceptions of the world, causing additional terror and fear of death. If readers believe stories to be exaggerated, however, sensationalism can have the opposite effect, causing distrust or desensitization [36].

## Managing uncertainty and fear: prediction and calls to action

A second, linked narrative concerned managing uncertainty and fear via prediction and calls to action. This represented an attempt to control or mitigate the impact or spread of the virus. Articles with an international focus conveyed the trope of foresight: using reports from the US, Italy and China to predict what was going to happen in the UK: what steps the UK would take, how many people would die, how people would die (e.g. 'alone'), which groups would suffer the highest death counts, and how the dead would be managed (e.g. mass graves, people having to bury their loved ones themselves).

Central to the narrative was creation of 'typical' and 'atypical' deaths and victims. From a TMT perspective, this introduced a pattern of predictability (and hence a sense of control). Such narratives can comfort a person who identifies with the 'atypical' group; while identification with the 'typical' group (for individuals or their loved ones) could promote fear.

Another attempt to manage the uncertainty and fear was through calls to action, offering ways of engaging in health behaviours to minimise the perceived threat of death [34]. We found that most calls for action focused on behaviour change and came from authority figures and people who had been bereaved, rather than from the traditional newspaper editorial. An exception was an article from The Sun which called for action to meet the needs of NHS staff who "begged" for PPE:

S.B.44: "the Government and the private sector must strain every sinew to get them the personal protective equipment they need before it's too late."

Most often, however, articles quoted bereaved relatives' social media posts about the death of their loved ones which urged people to adhere to social distancing. One article advised speaking to relatives and sharing end of life wishes:

Ti.B.38: Knowing that your family has an understanding of your wishes long before this stage would offer at least some comfort.

Individual deaths were used as warnings about the severity of the pandemic, such as:

TMi.A.12: A grief-stricken nephew has warned people to take coronavirus "very seriously" after his fit and healthy 60-year-old aunt fell asleep and never woke up

Using intimate personal stories to call for action may seem less prescriptive and be more persuasive than an editorial approach. From a TMT perspective, the media's use of bereaved people's voices may be a way of ascribing meaning to deaths (for example, serving as a warning to others), and offering symbolic immortality.

The narrative of managing uncertainty and fear—and the call to take the pandemic seriously—was associated with the use of bellicose language and war metaphors. COVID-19 was described as "this generation's Pearl Harbour" and the "enemy" (DM.B.34). From a TMT perspective, patriotic language encourages the public to embrace individual responsibility and putting needs of the UK before their own [10]. One article (S.B.26) described England's Chief Nurse, Ruth May, as "begging" the public to stay at home, quoting her requesting the public to respect the "sacrifices" of health workers. Therefore the article pitted people who were complying with social distancing regulations against those who were not, creating a battle against "lockdown rebels." Whilst the use of war metaphors may be effective for communicating the gravity of the situation, this practice can be criticised for unnecessarily amplifying fear, blaming those who died for "losing the fight" [37] and stoking nationalistic tensions [38, 39]. In data from Week B, there was less use of war metaphors, perhaps because of these criticisms and/or due to a perception that the "war" was being lost.

Other calls to action focused on bereaved people's requests for themselves (for example, for prayers) or braced the public with advice. For example, removing an unwell person's jewelry as a keepsake before their death to avoid further trauma ("I'm not allowed to have Dad's wedding ring or Star of David or neck chain as not able to retrieve it" (S.B.36)). Other advice prompted readers to "be strong" for their families in the months to come (DMi.A.6). Several articles referenced and linked to bereaved families' online donation pages, emphasising the financial strain experienced by bereaved people. Articles quoted bereaved celebrities' calls for people to donate to charities instead of sending flowers. There were also were explicit calls for the funeral industry to allow people to be able to attend in person or with technological adaptions. One article quoted the archbishop of Canterbury, Justin Welby, urging crematoriums and local authorities to respect the dead amid fears of mass graves and rushed burials:

DMa.B.49: "I say to crematoria and local authorities, don't just dispose of bodies like we did in the foot-and-mouth episode with cattle"

The message that more could be done to manage the uncertainty and/or impact of the pandemic on people (by individuals, organisations, and the government) was a common theme across all calls to action.

A stylistic feature of articles that included a call to action was a positive ending. Quotes from interviews or social media posts were used (DMa.A.8: "I think my father knew he was keeping everyone safe"). Others included extracts from speeches; for example, this quote from the Health Secretary Matt Hancock at a time when the UK death toll had reached 10,000:

> Me.B.39: "At the start of this crisis, people said that the NHS would be overwhelmed, and we've seen that and we've seen the risk of that elsewhere but not here. . . And that is because of the action that a huge number of people are taking and the incredible work of so many. There is more spare capacity now for critical care than there was when coronavirus first hit our shores."

## Mourning and loss

The third narrative focused on mourning the deaths caused by the pandemic, paying respect to the deceased and giving voice to the "outpouring of grief" associated with COVID-19. Fundamental to this narrative was telling the stories of the people who had died or had been bereaved. Articles which told stories of individual deaths typically included details of age, occupation and pre-COVID-19 health status:

> Gu.A.11: The otherwise fit 80-year-old then caught an infection in hospital

Aspects of the deceased's life or hobbies were mentioned as a way of honouring, memorialising and paying tribute.

Articles labelled and primed readers' emotions with adjectives such as "tragic." Tragedy was compounded by multiple losses, effects on family members, and the implied sacrifice of clinicians or key workers:

> S.B.57: Double agony NHS nurse fights for her life after her own mum dies of the deadly bug.

Articles described remarkable aspects of death, especially when these added to the perceived tragedy or sense of tragic irony: for example, contracting COVID-19 at a location which could not be avoided (such as at work); prior complaints made by deceased people about their working conditions including insufficient PPE; or the experience of multiple deaths in the same family. Including such detail signifies the newsworthiness of the deaths but can also be sensationalistic [5]. We found that the articles focused on tragic stories had a common structure outlining how COVID-19 challenged the grieving process: a person became unwell, loved ones were restricted or unable to visit in person; a person died with no loved ones present; the bereaved grieved alone due to social distancing; the bereaved were unable to uphold religious/cultural customs, such as washing the deceased's body, and encountered funeral restrictions. Relaying stories in this way framed COVID-19 bereavement as a series of disruptions that compounded the tragedy:

> DMa.A.43: The agony of saying goodbye to both the men in her life was compounded further by the fact she was unable to be with them during their final moments, unable to comfort either. Unable to say goodbye.

S.A.13: She wrote: "And now I start another complete quarantine, and think what kind of funeral I can plan from home, knowing it might not take place for quite awhile and might be a lot less than I think he deserves. More travesty!"

The narrative of mourning typically used euphemisms including "passed away" and "gone", language which distinguished these personal stories from the starkness of death counts and may indicate an attempt to be respectful and sensitive. The people who had died (particularly healthcare workers) were commonly referred to as "heroes" and "angels." Although death narratives were constructed as heroic throughout the pandemic, and this could be understood as a useful tool to support individual loss [40], some have criticised such euphemistic descriptions as preventing discussion about the limits of care and reciprocity in following public health guidance [41]. From a TMT perspective, references to those that had died or could be at risk as 'heroes' may bolster self-esteem by appealing to nationalistic identity, which offers symbolic immortality and hence protection from the terror of death.

## Omissions

In individual articles and the sample as a whole, there were no positive aspects to accounts of a COVID-19 death or the subsequent grief; for example, a death in which the wishes of the dying or their family members were satisfactorily met, or grief in which the bereaved person was well supported or able to find meaning and comfort despite social distancing restrictions. Nuanced accounts of how people with COVID-19 died, and the role of symptom management or specialist palliative care in the dying process, were also lacking. By implication, all deaths from COVID-19 were portrayed as 'bad' deaths and all experiences of bereavement and funerals at this time as wholly negative. There was also limited practical advice about what to do if your loved one became seriously ill, was admitted to hospital or died.

Some articles described positive acts, such as a Metro piece (Me.1.16) reporting that a man had asked the public to send cards to his bereaved grandmother to help her feel less alone while family could not see her. Another article in The Metro gave guidance from other bereaved people regarding what to say to someone who is grieving:

Me.B.46: If you know someone who has lost a loved one—either due to COVID-19 or another reason—it can be very difficult to know what the 'right' thing is to say to them, or if you should say anything at all, especially if you have never been in this type of situation before.

However, these were notable exceptions. According to the discourse, there seemed few positive or supportive actions that bereaved people, or those supporting them, could take.

## Discussion

The COVID-19 pandemic has brought human mortality into stark relief, and this is reflected in media reportage. In this analysis of UK online newspapers' coverage of bereavement due to COVID-19 during the first wave of the pandemic, we identified three main narratives at play: fear of an uncontrollable and unknown virus; managing uncertainty and fear; and mourning and loss. These narratives reflect a journalistic tension between maximizing and minimising readers' fear, as well as a cultural attempt to make sense of a new and evolving situation.

COVID-19 was presented in online newspapers in the UK at this time as an uncontrollable, unpredictable virus about which little was known. Sensationalist language, particularly in article headlines and introductions, drove home its deadly nature and emphasised its

risks, with articles typically focusing on unusual or especially distressing COVID-19 deaths. At the same time, uncertainty and fear were managed by presenting some deaths as expected or more likely to occur (creating categories of 'typical' and 'atypical' deaths), as well as predicting what might happen as the virus unfolded, and galvanising calls to action. The rolling 7-day death rate of people aged 40 or under remained below 1 per 100,000 [42], yet deaths among young people were a focus of news reportage and portrayed as particularly tragic. As well as these 'atypical' deaths, the media focussed on tragic cases where deaths more 'typical' or expected (e.g. among clinicians at a higher risk), particularly where there was a sense of irony in the death (e.g. older people with comorbidities who had lived with a particular condition for many years).

A third narrative focused on telling the stories of those who died and those who had been bereaved, giving voice to collective and personal grief. This story-telling often used emotive and euphemistic language, framing bereavement as a series of compounded losses which increased the pain of grieving and the tragedy of the death.

News media mediate people's experiences of illness, death and grief by providing opportunities and context to imagine how their experiences fit in with a wider community [43]. Central to the media narrative we identified was the use of sensationalism, which is common in media reporting on deaths [5]. While the large-scale impact of COVID is of course sensational in the context of typical British news-cycles, and sensationalist language can arguably enable journalists to reflect the moral interests of the community [44], sensationalism in this context was not often balanced by clear reporting of facts. This is exemplified by the common use of the word "ban" to describe what was, in reality, a nuanced and varied set of restrictions across different areas of healthcare and funeral care. In an accelerating and ever-more competitive news-media market, sensationalism has become an increasingly prevalent tool for combining the news's function as a source of information with the attention-grabbing excitement of a shocking headline or story [36]. As the competition for human attention intensifies and newspapers compete for survival in an increasingly competitive market, sensationalism might be to a degree inevitable. However, where sensationalist COVID-related reportage lacked the appropriate balance and focused too strongly on shocking headlines, it may have contributed towards public confusion regarding what was and what was not possible under COVID-19-related regulations and recommendations [45]. Sensationalism also stokes fear and anxiety among readers, which can cause them to disengage [46], and might be particularly detrimental for those already bereaved by COVID-19 [47]. Reporting the daily death tolls, another common trope, may also have served to increase fear (and raise readership numbers) initially, but can lead to desensitisation, as seen in Lebanon after increased exposure to terrorism-related news [48]. It therefore comes as little surprise that in a UK survey of people bereaved since March 2020 (n = 503), 62% felt that the public has become desensitised to death due to the way in which the COVID-19 pandemic was documented [49].

We also found important omissions in how bereavement due to COVID-19 was reported: a lack of nuance in accounts of death, dying and bereavement, and little discussion of practical ways to support those seriously ill or bereaved. COVID-19 deaths were portrayed as inherently 'bad', leading to an unnatural kind of grief in which the needs of bereaved people were rarely, if ever, met. The homogeneity in these accounts leaves little room for alternative experiences or perspectives. A lack of attention to practical steps that someone can take leaves readers fearful and disempowered. Similarly, a focus on what is impossible due to funeral restrictions could be tempered with a description of what is still possible and could be of benefit. Evidence suggests that in fact funeral restrictions do not inevitably lead to poor bereavement outcomes [50]; rather, the value of post-death rituals to the bereaved lies in their personal meaningfulness and ability to demonstrate social support. The innovations of funeral officiants and those

bereaved at this time are testament to the possibility of meaningful services and acts of memorialisation.

Journalists and editors have played a critical role in the pandemic; figures indicate that all the news sites we included in our analysis saw a monthly increase in the average amount of time adult visitors spent on their sites from December 2019 to March 2020 [51]. Reflecting the widespread uncertainty at the beginning of the pandemic in the UK, the media attempted to rapidly make sense of COVID-19 and its impact, with a particular emphasis on what might happen next, who was most at risk of dying, the context of their deaths and the loss of cultural rituals. Alongside this sense-making, the media conveyed behaviour change messages and epidemiological and political news. The impact of playing these roles in the midst of a crisis both global and personal should not be under-estimated [52]. Duncan [53] notes how journalists can experience vicarious trauma from their work and become desensitised, leaving them with a negative worldview. There can also be an emotional dissonance between editorial expectations and what journalists feel. To manage their emotional labour, journalists may focus on the process of researching and writing rather than the content, or seek to protect their audiences by processing content to keep the most disturbing aspects out of the public eye.

This study has implications for how health-related and death-related news is communicated and how the media response to deaths could be improved in this and future mass bereavement events. There appears to be a delicate balance required in using fear-based narratives that can help generate action and solidarity without increasing fear in a way that may exacerbate the situation or negatively affect bereaved people. As noted in relation to other 'sensationalist' topics and suicide, the media have a role to play in nuanced reporting [44, 54, 55]. As in the H1N1 pandemic [10] and the reporting of other diseases such as cancer [36, 56], we found bellicose language was commonly used, and whilst we noticed a reduction in the use of war metaphors over time, the continued use of metaphors that presented healthcare workers as heroes or angels can make it difficult to see them as human beings with needs and fears. Within the context of COVID-19, our analysis indicates that there is scope for news reports to not present all unknowns as immediately threat-inducing. More news coverage could highlight how people are grieving, managing and seeking stability—both to help people understand how they and others might experience grief and to provide advice on how to support others who are bereaved. Our research was conducted in the first wave of the COVID-19 pandemic in the UK, and as many countries continue to face additional waves of the pandemic and news coverage evolves, these findings are particularly pertinent for ongoing reporting.

There are several strengths to this study, including the variety of newspapers included; the use of clearly defined time-frames that could be compared and contextualised with other events; the iterative, team-based and reflexive analysis; and the application of TMT in interpreting the findings. We limited our analysis to UK online news media and focused our analysis on a one-month period of grief and bereavement due to COVID-19. A wider or different scope might have changed our findings. We also focused only on the language element of discourse and did not consider the placing and prominence of the articles (either in the newspaper or on the website) and how that might influence their effect [57].

We would like to highlight three areas for future research. First, research into the effects of media representations of death and bereavement on the public (particularly those bereaved by COVID-19) would be highly valuable to help guide future reporting. Second, research is needed to investigate how media coverage (both newspapers and television) has evolved during the course of the pandemic, to understand whether and how the framing we identified in March-April 2020 has changed over time. Third, we found that social media posts from bereaved people were used as the basis of many articles, perhaps due to a lack of interviewees willing to talk about their experiences of death and bereavement, or because many people took

to social media at this time [58]. Although inclusion of social media might be considered a more public-led approach to journalism, content from personal social media is selected using gatekeeping techniques, making it typically consistent with traditional media narratives rather than representative of a more wide-ranging and democratic discourse [59]. A key area for future research is how social media has reinforced and/or challenged the cultural messaging conveyed by the written press, and the implications of this, particularly for those population groups which use social media channels more than news organisations to access information [60].

## Conclusion

Online UK newspapers' reporting of bereavement due to COVID-19 in spring 2020 reflected three main narratives: (1) fear of an uncontrollable, unknown new virus and its uncertain consequences—associated with sensationalist language in place of clear reportage and a sense of helplessness and confusion; (2) managing uncertainty and fear via prediction of the future and calls for behaviour change, associated with use of war metaphors; and (3) mourning and loss narratives that paid respect to the deceased and gave voice to grief, associated with euphemistic or glorifying language. At this early stage of the pandemic, there was an evident tension between discourse which emphasised the existential threat of a novel disease and discourse which attempted to mitigate or control that threat. There was a lack of nuanced accounts of death and grief and limited practical advice about what to do if a loved one became seriously ill or died, with bereavement due to COVID-19 presented as a series of tragedies. While the impact of this reporting on the public is unknown, less homogenous and fear-based coverage, alongside clear information and signposting to support, could better serve the public, especially those bereaved by COVID-19.

## Supporting information

**S1 Table. Number of included articles by date, contextualised with total deaths to date and key events in the UK.**
(DOCX)

## Author Contributions

**Conceptualization:** Ryann Sowden, Lucy E. Selman.

**Formal analysis:** Ryann Sowden, Erica Borgstrom, Lucy E. Selman.

**Investigation:** Ryann Sowden.

**Methodology:** Ryann Sowden, Erica Borgstrom, Lucy E. Selman.

**Project administration:** Ryann Sowden.

**Supervision:** Lucy E. Selman.

**Writing – original draft:** Ryann Sowden, Lucy E. Selman.

**Writing – review & editing:** Erica Borgstrom, Lucy E. Selman.

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
