## [Decision Letter · Decision Letter 0]

5 Feb 2021

PONE-D-20-39786

‘It's like being in a war with an invisible enemy’: A document analysis of bereavement due to COVID-19 in UK newspapers

PLOS ONE

Dear Dr. Selman,

Thank you for submitting your manuscript to PLOS ONE. After careful consideration, we feel that it has merit but does not fully meet PLOS ONE’s publication criteria as it currently stands. Therefore, we invite you to submit a revised version of the manuscript that addresses the points raised during the review process.

We look forward to receiving your revised manuscript.

Kind regards,

Johnson Chun-Sing Cheung, D.S.W.

Academic Editor

PLOS ONE

**Comments to the Author**

Reviewer #1: This is a valuable and timely analysis of press coverage of grief and loss during the Covid-19 pandemic, focusing on leading online news outlets from/about the UK. The analysis is sound and the conclusions drawn are appropriate. My main concern is with the framing of the study and some of the discussion and interpretations. Fundamentally, the analysis repeatedly critiques the news' outlets "sensationalism." The theoretical framework of terror management theory helps all of this hang together. But keep in mind what the purpose of journalism is. I'm reminded of a famous newspaper editor's quip. "It's our job to report the planes that crash. It's not our job to report all the planes that land safely." (I'm not positive, but I think this is the famous Ben Bradlee of the Washington Post.)

This analysis would help by acknowledging this point clearly at various points -- rewriting a few sentences here and there, and thinking carefully as you re-read the conclusions. Sensationalism is obviously increasing across all of journalism -- especially 'mainstream' journalism -- as the competition for human attention intensifies. Along those lines, mention how journalism is evolving at an accelerating pace even as Covid-19 is evolving at a pace that epidemiologists are rushing to keep up with. This really struck me in the first paragraph of the 'Discussion' section. "As the virus unfolded," we see a speedy evolution in information about the virus (and conspiracies!). This is why we see so much attention to the "infodemic" that is spreading along with the "pandemic." The use of the term "evolution" here is not a casual metaphor: I am thinking of influential interpretations of journalism through evolutionary lenses -- the Chicago School Sociologist Robert Park taught a course in the 1920s, "A Natural History of the Newspaper," that was part of the paradigmatic application of evolutionary metaphors to discourse and news. Today we're seeing all of this unfold at a speedy pace, driven by the massive volume of information -- all of which 'traditional' mainstream news outlets must now compete with. This creates massive, exponential possibilities in the mathematics of combinatorics -- snippets of information, data, images, metaphors, etc. -- that circulate online in velocities that generate powerful memes. Again, the word is not casual: trace 'meme' back to its original source (Dawkins in 1976) and then look at how the linguist Noam Chomsky analyses the circulation of various discourses today...

Sorry for the rant. I think this is a great paper. I just think we need to be careful about expecting 'nuance' from an industry and profession that is governed, like it or not, but other fundamental imperatives for its survival.

Small issues to correct before publication. I am a bit confused on where the figure of 5 people directly bereaved comes from; is this from some kind of estimate? Or just an assumption to make a point? If it's the latter, envision and describe a bell curve (kurtosis and skew unknown) of different deaths touching different numbers of people. Even if the mean is the same at 5, there will be enormous variation in how bereavement cascades through a population, just as Covid-19 is cascading through global and local populations.

I am a bit confused on what is meant by 'non-bereavement-related grief and why it's excluded.

A few minor typos throughout the manuscript. Extra common after metaphors on p. 3, four lines from bottom; replace colon with long dash in middle paragraph; replace colon with long dash on line 2 p. 5; missing "as" after TMT on page 8 three lines from bottom; phenomena on line 1 page 9; extra reflect in middle of p. 9.; readers' vs. reader's on page 15 three lines from bottom; missing 'by' within quote from DMA.A.43; missing "were" in first line below 'Omissions'; extra period page 20 three lines from bottom;

*

All in all, a valuable contribution!

Reviewer #2: This well-written manuscript explores media portrayals of COVID-19 bereavements in the UK during the earliest stages of the pandemic. The manuscript uses a broad range of media sources, and situates their framing and discussion using Terror Management Theory. Despite these strengths, I have several suggestions for enhancing the manuscript’s clarity, precision, and potential impact.

1. it’s important to state very directly upfront that the study takes place during the earliest stages of the pandemic, when levels of uncertainty were high. I would be very curious to see if portrayals evolved throughout the course of the pandemic, or whether different approaches were taken once rates started to dip. The authors could do more to discuss the extent to which their results reflect the particular historical moment in the pandemic.

2. A very minor point. The opening paragraphs states that we can presume 5 people are directly affected by each COVID-19 death. Where did this ratio come from? I would suggest consulting and citing work by Ashton Verdery, which constructed a COVID-1 bereavement multiplier.

3. I would suggest providing a line or two about the demographic or market targeted by each of the newspaper included in the study, and perhaps a sense of what share of the population reads them. This may help to understand how each frames and targets their coverage.

4. You may want to list a priori the types of words/phrases you would expect to see as ‘evidence’ in support of a framing that draws on TMT themes.

5. The methods are well-described, although I’d like to learn more about inter-rater reliability in the coding/classifying.

6. I wonder whether the language used to describe victims is unique to COVID or whether similar language is used for other widespread conditions like cancer or deaths due to addiction.

7. It’s not surprising that newspapers focus on the odd or atypical case, like youthful deaths from COVID. News is “man bites dog” not “dog bites man.” One way to contextualize your results may be to provide some data to demonstrate how atypically youthful deaths are, for instance. Providing some quantitative information on the nature of COVID-deaths, including both the characteristics of those afflicted and who die, as well as data on indicators of ‘good’ and ‘bad’ deaths (e.g., duration of condition, on ventilator, etc.” would help to strengthen some of the paper’s claims.

8. The discussion of managing uncertainty/behavior change might benefit from some attention to timeline. When was the first case and death in the UK, relative to the US, Italy, and China, for instance. It would help to contextualize the findings against a background of information spread/knowledge.

9. The authors critique the print media on the grounds that they do not have enough uplifting messages in their coverage. Perhaps this kind of ‘feel good’ coverage is more likely part of TV rather than print?

10. It may be beyond the scope of this manuscript, but I would love to see even a brief mention of how media coverage has changed in the months since the initial analyses were done, to provide insights into whether the particular framing in March/April 2020 reflected the true uncertainty regarding this new virus.

I hope these comments are helpful as the author(s) revise their work.

---

## [Author Response · Author response to Decision Letter 0]

13 Feb 2021

13 February 2021

Dear Editors, 

Re: ‘It's like being in a war with an invisible enemy’: A document analysis of bereavement due to COVID-19 in UK newspapers (PONE-D-20-39786)

Many thanks for the reviewers’ comments on the above manuscript, and the opportunity to revise and resubmit. We are pleased to enclosed our revised manuscript and a response to the reviewers’ comments. 

With best wishes, 

Dr Lucy Selman, Dr Ryann Sowden and Dr Erica Borgstrom 

Journal requirements 

Thank you, we have ensured these requirements are met. 

Reviewer 1

This is a valuable and timely analysis of press coverage of grief and loss during the Covid-19 pandemic, focusing on leading online news outlets from/about the UK. The analysis is sound and the conclusions drawn are appropriate. 

Many thanks 

My main concern is with the framing of the study and some of the discussion and interpretations. Fundamentally, the analysis repeatedly critiques the news' outlets "sensationalism." The theoretical framework of terror management theory helps all of this hang together. But keep in mind what the purpose of journalism is. I'm reminded of a famous newspaper editor's quip. "It's our job to report the planes that crash. It's not our job to report all the planes that land safely." (I'm not positive, but I think this is the famous Ben Bradlee of the Washington Post.) This analysis would help by acknowledging this point clearly at various points -- rewriting a few sentences here and there, and thinking carefully as you re-read the conclusions. Sensationalism is obviously increasing across all of journalism -- especially 'mainstream' journalism -- as the competition for human attention intensifies. Along those lines, mention how journalism is evolving at an accelerating pace even as Covid-19 is evolving at a pace that epidemiologists are rushing to keep up with. This really struck me in the first paragraph of the 'Discussion' section. "As the virus unfolded," we see a speedy evolution in information about the virus (and conspiracies!). This is why we see so much attention to the "infodemic" that is spreading along with the "pandemic." The use of the term "evolution" here is not a casual metaphor: I am thinking of influential interpretations of journalism through evolutionary lenses -- the Chicago School Sociologist Robert Park taught a course in the 1920s, "A Natural History of the Newspaper," that was part of the paradigmatic application of evolutionary metaphors to discourse and news. Today we're seeing all of this unfold at a speedy pace, driven by the massive volume of information -- all of which 'traditional' mainstream news outlets must now compete with. This creates massive, exponential possibilities in the mathematics of combinatorics -- snippets of information, data, images, metaphors, etc. -- that circulate online in velocities that generate powerful memes. Again, the word is not casual: trace 'meme' back to its original source (Dawkins in 1976) and then look at how the linguist Noam Chomsky analyses the circulation of various discourses today...

Thank you for these interesting and helpful comments. We recognise that the purpose of journalism is not (just) to report facts clearly and/or to be helpful or supportive, but to sell newspapers and engage readers. We have added a section to the Discussion to contextualise sensationalism and provide more nuance, as follows: 

“While the large-scale impact of COVID is of course sensational in the context of typical British news-cycles, and sensationalist language can arguably enable journalists to reflect the moral interests of the community(43), sensationalism in this context was not often balanced by clear reporting of facts. This is exemplified by the common use of the word “ban” to describe what was, in reality, a nuanced and varied set of restrictions across different areas of healthcare and funeral care. In an accelerating and ever-more competitive news-media market, sensationalism has become an increasingly prevalent tool for combining the news’s function as a source of information with the attention-grabbing excitement of a shocking headline or story(36). As the competition for human attention intensifies and newspapers compete for survival in an increasingly competitive market, sensationalism might be to a degree inevitable. However, where sensationalist COVID-related reportage lacked the appropriate balance and focused too strongly on shocking headlines, it may have contributed towards public confusion regarding what was and what was not possible under COVID-19-related regulations and recommendations(44).”

Sorry for the rant. I think this is a great paper. I just think we need to be careful about expecting 'nuance' from an industry and profession that is governed, like it or not, but other fundamental imperatives for its survival. 

Thank you – and we appreciate your suggestions for improving the above discussion of sensationalism. 

Small issues to correct before publication. I am a bit confused on where the figure of 5 people directly bereaved comes from; is this from some kind of estimate? Or just an assumption to make a point? If it's the latter, envision and describe a bell curve (kurtosis and skew unknown) of different deaths touching different numbers of people. Even if the mean is the same at 5, there will be enormous variation in how bereavement cascades through a population, just as Covid-19 is cascading through global and local populations. Thank you for highlighting the likely variation in how bereavement affects a population, which we now include on page 3. 

The reference of 5 bereavements per death comes from pre-COVID work by Katherine Shear on complex grief. However, reviewer 2 highlighted a helpful new research article by Verdery et al. which identified each COVID-19 death as leaving 9 people bereaved. We now use this metric and reference it accordingly: 

“While there are likely to be large variations in how bereavement affects a population, recent research in the USA estimates that each death from COVID-19 leaves 9 people bereaved(4). Using this metric, an estimated 1.08 million people in the UK are currently grieving a COVID-19 death.”

I am a bit confused on what is meant by 'non-bereavement-related grief and why it's excluded. 

For clarity, we have now changed this section to: 

“…articles discussing grief unrelated to death (e.g. grieving loss of social contact), or a non-human death (e.g. of a pet) as this was outside the focus of the study.”

A few minor typos throughout the manuscript. Extra common after metaphors on p. 3, four lines from bottom; replace colon with long dash in middle paragraph; replace colon with long dash on line 2 p. 5; missing "as" after TMT on page 8 three lines from bottom; phenomena on line 1 page 9; extra reflect in middle of p. 9.; readers' vs. reader's on page 15 three lines from bottom; missing 'by' within quote from DMA.A.43; missing "were" in first line below 'Omissions'; extra period page 20 three lines from bottom; Thank you – we have corrected these typos. 

All in all, a valuable contribution! 

Thank you

Reviewer 2

This well-written manuscript explores media portrayals of COVID-19 bereavements in the UK during the earliest stages of the pandemic. The manuscript uses a broad range of media sources, and situates their framing and discussion using Terror Management Theory. Despite these strengths, I have several suggestions for enhancing the manuscript’s clarity, precision, and potential impact. 

Thank you for your praise of the paper and your helpful suggestions to strengthen it. 

1. it’s important to state very directly upfront that the study takes place during the earliest stages of the pandemic, when levels of uncertainty were high. I would be very curious to see if portrayals evolved throughout the course of the pandemic, or whether different approaches were taken once rates started to dip. The authors could do more to discuss the extent to which their results reflect the particular historical moment in the pandemic. 

We did already contextualise our findings, e.g. in supplementary table 1 we present the dates of key events in relation to the articles we analysed, and we state in the Discussion:

“Reflecting the widespread uncertainty at the beginning of the pandemic in the UK, the media attempted to rapidly make sense of COVID-19 and its impact, with a particular emphasis on what might happen next, who was most at risk of dying, the context of their deaths and the loss of cultural rituals. Alongside this sense-making, the media conveyed behaviour change messages and epidemiological and political news.” 

However, to further emphasise chronology we have now added further mention of the context on pages 19 and 22 of the Discussion. 

2. A very minor point. The opening paragraphs states that we can presume 5 people are directly affected by each COVID-19 death. Where did this ratio come from? I would suggest consulting and citing work by Ashton Verdery, which constructed a COVID-1 bereavement multiplier.

Thank you for highlighting this useful paper, which we now reference in the Introduction:

“While there are likely to be large variations in how bereavement affects a population, recent research in the USA estimates that each death from COVID-19 leaves 9 people bereaved(4). Using this metric, an estimated 1.08 million people in the UK are currently grieving a COVID-19 death.”

3. I would suggest providing a line or two about the demographic or market targeted by each of the newspaper included in the study, and perhaps a sense of what share of the population reads them. This may help to understand how each frames and targets their coverage.

Thank you – we have now added this information on page 5. 

4. You may want to list a priori the types of words/phrases you would expect to see as ‘evidence’ in support of a framing that draws on TMT themes.

We used TMT as a theoretical lens because it complemented and enriched the initial codes developed in our content analysis, providing an appropriate perspective through which to interpret and explore the data. We did not specify a priori how we would use TMT or judge its applicability – this was determined through iterative coding and consideration of relevant theories in team discussions. We have added a phrase further specifying why TMT complemented the analysis:

 “We selected TMT as it complemented many codes identified in the initial content analysis (for example, related to the effects of funeral restrictions on the bereaved and the evident fear and control in reportage), and provides a useful analytical perspective on the themes and tropes related to newspapers’ sense-making during the COVID-19 pandemic.” (page 9)

5. The methods are well-described, although I’d like to learn more about inter-rater reliability in the coding/classifying. We used a team approach to the development of the coding framework, the consideration of new themes and emerging findings and the application of TMT. As described in the Methods (and now further highlighted in the text), one person applied the coding frame, so there was no inter-rater reliability to assess:

“To develop a coding framework, three varied and rich articles were selected by RS from the Week A dataset. Each co-author independently read these articles, making notes on latent and manifest themes. The coding and notes were discussed as a group, and organised to develop a hierarchical coding frame of defined themes and sub-themes for the first week (LS). This process was repeated for the Week B dataset. These coding frames were combined, refined and expanded to capture the themes and salient issues from both weeks. RS applied the coding frame to all included articles, with regular meetings with LS and EB to discuss narrative formation within articles, themes within and across each week, and social theories that could help us understand the discourses. We reflected on our re-reading of the articles as the pandemic unfolded and how our interpretations could shift over time due to this. Additional emergent themes (not identified in the initial coding framework) were discussed, added as needed, and applied consistently across the dataset by RS.” (page 8)

6. I wonder whether the language used to describe victims is unique to COVID or whether similar language is used for other widespread conditions like cancer or deaths due to addiction. 

We now reference in the Discussion the use of such metaphors in other diseases such as cancer (page 21).

7. It’s not surprising that newspapers focus on the odd or atypical case, like youthful deaths from COVID. News is “man bites dog” not “dog bites man.” One way to contextualize your results may be to provide some data to demonstrate how atypically youthful deaths are, for instance. Providing some quantitative information on the nature of COVID-deaths, including both the characteristics of those afflicted and who die, as well as data on indicators of ‘good’ and ‘bad’ deaths (e.g., duration of condition, on ventilator, etc.” would help to strengthen some of the paper’s claims.

Thanks for this interesting suggestion. We have revised and added some more contextual details to the Discussion in response and to strengthen our claims (see below). However, it is worth noting that full data on the nature of COVID-19 deaths is not yet widely known – more research in this area is needed. 

“COVID-19 was presented in online newspapers in the UK at this time as an uncontrollable, unpredictable virus about which little was known. Sensationalist language, particularly in article headlines and introductions, drove home its deadly nature and emphasised its risks, with articles typically focusing on unusual or especially distressing COVID-19 deaths. At the same time, uncertainty and fear were managed by presenting some deaths as expected or more likely to occur (creating categories of ‘typical’ and ‘atypical’ deaths), as well as predicting what might happen as the virus unfolded, and galvanising calls to action. The rolling 7-day death rate of people aged 40 or under remained below 1 per 100,000(42), yet deaths among young people were a focus of news reportage and portrayed as particularly tragic. As well as these ‘atypical’ deaths, the media focussed on tragic cases where deaths more ‘typical’ or expected (e.g. among clinicians at a higher risk), particularly where there was a sense of irony in the death (e.g. older people with comorbidities who had lived with a particular condition for many years).” 

8. The discussion of managing uncertainty/behavior change might benefit from some attention to timeline. When was the first case and death in the UK, relative to the US, Italy, and China, for instance. It would help to contextualize the findings against a background of information spread/knowledge.

We contextualise the data in supplementary table 1 against key dates of the COVID-19 pandemic in the UK. We have now added to the table some of the reviewer’s additional international suggestions – thank you. 

9. The authors critique the print media on the grounds that they do not have enough uplifting messages in their coverage. Perhaps this kind of ‘feel good’ coverage is more likely part of TV rather than print?

 While this is an interesting idea, we are not aware of any research showing this. There are several examples of ‘good news’ newspapers and online sites that indicate a desire for this kind of positive print media.

10. It may be beyond the scope of this manuscript, but I would love to see even a brief mention of how media coverage has changed in the months since the initial analyses were done, to provide insights into whether the particular framing in March/April 2020 reflected the true uncertainty regarding this new virus. We found there was a great deal of uncertainty in reporting during the first wave in the UK, but that this was often also associated with fear-based coverage and sensationalism; even at the beginning of the pandemic, uncertainty did not have to be presented in this way. 

Discussing how media reporting might have changed in subsequent months is indeed outside the remit this paper. While we could hypothesise about possible changes, it would feel like speculation without formal analysis. For example, given the volume of media on COVID-19 death and bereavement, it would be cherry-picking to select a few examples of more positive or helpful coverage in more recent months. However, we agree this is an important area of future research and have added it on page 22:

 “Second, research is needed to investigate how media coverage (both newspapers and television) has evolved during the course of the pandemic, to understand whether and how the framing we identified in March-April 2020 has changed over time.”

---

## [Editor Report · Decision Letter 1]

17 Feb 2021

‘It's like being in a war with an invisible enemy’: A document analysis of bereavement due to COVID-19 in UK newspapers

PONE-D-20-39786R1

Dear Dr. Selman,

We’re pleased to inform you that your manuscript has been judged scientifically suitable for publication and will be formally accepted for publication once it meets all outstanding technical requirements.

Kind regards,

Johnson Chun-Sing Cheung, D.S.W.

Academic Editor

PLOS ONE

---

## [Editor Report · Acceptance letter]

24 Feb 2021

PONE-D-20-39786R1 

‘It's like being in a war with an invisible enemy’: A document analysis of bereavement due to COVID-19 in UK newspapers 

Dear Dr. Selman:

I'm pleased to inform you that your manuscript has been deemed suitable for publication in PLOS ONE. Congratulations! Your manuscript is now with our production department. 

Kind regards, 

on behalf of

Dr. Johnson Chun-Sing Cheung 

Academic Editor

PLOS ONE